# Fusion-Based Versatile Video Coding Intra Prediction Algorithm with Template Matching and Linear Prediction

**DOI:** 10.3390/s22165977

**Published:** 2022-08-10

**Authors:** Dan Luo, Shuhua Xiong, Chao Ren, Raymond Edward Sheriff, Xiaohai He

**Affiliations:** 1College of Electronics and Information Engineering, Sichuan University, No. 24 South Section 1, Yihuan Road, Chengdu 610065, China; 2Department of Computer Science, Edge Hill University, Ormskirk L39 4QP, UK

**Keywords:** Versatile Video Coding, intra prediction, template matching, linear prediction

## Abstract

The new generation video coding standard Versatile Video Coding (VVC) has adopted many novel technologies to improve compression performance, and consequently, remarkable results have been achieved. In practical applications, less data, in terms of bitrate, would reduce the burden of the sensors and improve their performance. Hence, to further enhance the intra compression performance of VVC, we propose a fusion-based intra prediction algorithm in this paper. Specifically, to better predict areas with similar texture information, we propose a fusion-based adaptive template matching method, which directly takes the error between reference and objective templates into account. Furthermore, to better utilize the correlation between reference pixels and the pixels to be predicted, we propose a fusion-based linear prediction method, which can compensate for the deficiency of single linear prediction. We implemented our algorithm on top of the VVC Test Model (VTM) 9.1. When compared with the VVC, our proposed fusion-based algorithm saves a bitrate of 0.89%, 0.84%, and 0.90% on average for the Y, Cb, and Cr components, respectively. In addition, when compared with some other existing works, our algorithm showed superior performance in bitrate savings.

## 1. Introduction

In recent years, Internet traffic comprising multimedia data has increased rapidly, with 80% expected to be of video content in the coming years. Therefore, efficient video compression is very important. The new generation video coding standard Versatile Video Coding (VVC) [1] was officially released in July 2020. On the basis of the High Efficient Video Coding (HEVC) [2] standard, VVC has introduced many advanced technologies to further improve compression performance. Simultaneously, VVC supports a wide variety of video types such as High Definition (HD) and Ultra HD (UHD) resolution, Wide Color Gamut (WCG) video sequences, Virtual Reality (VR) videos, ultra-low delay applications, and so on.

Currently, VVC continues to employ a block-based hybrid coding framework, while most of its coding tools have been improved. In terms of block division, in addition to the traditional Quad Tree (QT), to make the partitioning results more suitable for video content, Binary Tree (BT) and Ternary Tree (TT) [3] have also been introduced. For intra prediction, the angle prediction modes are more detailed, and Position Dependent Intra Prediction Combination (PDPC) [4,5], Multiple Reference Line (MRL) [6,7], Intra Sub-Partition (ISP) [8,9], Cross Component Linear Model (CCLM) [10,11,12], Matrix Weighted Intra Prediction (MIP) [13,14] and Wide-Angle Intra Prediction (WAIP) [15,16,17] have been utilized to optimize intra prediction performance. Bi-Directional Optical Flow (BDOF) [18,19], Decoder-side Motion Vector Refinement (DMVR) [20], and Affine Motion Compensation (AMC) [21,22] have been used to improve inter prediction precision. Multiple Transform Selection (MTS) [23], Sub-Block Transform (SBT) [24], and Low Frequency Non-Separable Transform (LFNST) [25] have been employed to further eliminate frequency redundancy. In addition, many other modules have also been optimized.

Intra prediction predicts the current pixels by using the reconstructed pixels of the current frame to remove spatial redundancy. Recently, many works on intra prediction have been performed. Schneider et al. [26] proposed an algorithm named Sparse Coding-Based Intra Prediction (SCIP) to further improve the performance of VVC. To better predict the areas with complex textures, Li et al. [27] presented an improved intra prediction mode combination method and introduced an efficient mode coding method of syntax elements to enhance the coding performance. Yoon et al. [28] designed a method to obtain the parameters of CCLM more precisely, which compensated for the coding loss of the simplified CCLM mode. To fully utilize the advantages of Intra Block Copy (IBC) and palette coding, Zhu et al. [29] designed a compound palette mode to improve the performance of VVC on screen content coding. Since IBC cannot deal well with geometric transformations, Jayasingam et al. [30] extended IBC to adapt contents with zooms, rotations, and stretches. In [31], Weng et al. presented an L-shape-based iterative algorithm to improve intra prediction accuracy, and a residual median edge detection method was also proposed to address edge information. In [32], Yoon et al. exploited the number of occurrences of the modes in the neighboring blocks to extend the Most Probable Mode (MPM). Wang et al. [33] designed a Sample Adaptive Offset (SAO) acceleration method to reduce the complexity of VVC. Saha et al. [34] analyzed the decoder complexity of VVC on two different platforms.

Intra prediction plays an important role in video coding, and effectively compressing data can reduce the workload of the sensors. Although there are already some works that have achieved good results, there have been few that have taken the different features of different blocks and different modes into account. Hence, there is still room for further improvement. In this paper, we propose a fusion-based algorithm. The main contributions of our work are as follows:(1)We proposed a fusion-based adaptive template matching method. Its core idea is adaptive template matching. This method sufficiently considers the influence of matching errors on prediction results. The mode using this method is named mode 67;(2)We designed a fusion-based linear prediction method. Its core idea is linear prediction. This method fully considers the linear relationship between reference pixels and the pixels to be predicted, and the correlation between different models. The mode utilizing this method is named mode 68.

The remainder of the paper is organized as follows. Section 2 introduces the related works. In Section 3, our proposed fusion-based algorithm is presented in detail. In Section 4, we conduct some experiments to verify the effectiveness of our proposed algorithm, and conclusions are drawn in Section 5.

## 2. Related Works

The Template Matching Prediction (TMP) algorithm is a widely used method in video coding. The TMP algorithm takes the reference pixels of the current Prediction Unit (PU) block as a reference template. In the reconstructed area, a certain criterion is employed to select some candidate templates with the least error, from which the final prediction value is obtained. To reduce the effect of compression noise, Tan et al. [35] proposed using the average value of several candidate templates. Gayathri et al. [36] presented a region-based TMP algorithm that could reduce the complexity and obtain good performance. In [37], Gayathri et al. further decreased the memory requirements and number of computations at the decoder side. Considering that only the local reference samples could not deal well with complex areas, Lei et al. [38] designed a two-step progressive method to use both local (derived by the high frequency coefficients) and non-local information (obtained through TMP). These methods achieved good performance, however, simply averaging does not take the different importance of different blocks into account, and a linear approach is not adaptive and does not directly consider the error. To better predict PUs with rich texture information, we proposes a fusion-based adaptive template matching method in this paper.

Linear prediction assumes a linear relationship between reference samples and the samples to be predicted. The final prediction value is obtained by constructing a linear function. Typically, CCLM linearly obtains the chroma prediction value through the corresponding reconstructed luminance samples. The process of CCLM is shown in Figure 1. First, the co-located luminance block is down-sampled. Then, the reference pixels of the current chroma block and the down-sampled luminance block are used to calculate the parameters by constructing a linear function. Finally, the chroma prediction value is obtained by a linear function.

As shown in Equation (1), $pred_{c}\left(i,j\right)$ is the chroma pixel to be predicted, and $rec_{L}^{’}\left(i,j\right)$ denotes the down-sampled reconstructed luma pixel. (i,j) is the position and α, β are the parameters.
(1)$$pred_{c}\left(i,j\right)=\alpha \times rec_{L}^{’}\left(i,j\right)+\beta$$

Many related works have been undertaken based on CCLM. Ghaznavi-Youvalari et al. [39] merged CCLM with an angle mode derived from the corresponding luma block to improve the chroma prediction accuracy. Zhang et al. [40] introduced three methods including Multi-Model CCLM (MMCCLM), Multi-Filter CCLM (MFCCLM), and linear mode angle prediction to further enhance the coding efficiency of CCLM. However, there have been few linear prediction optimizations for the luminance component. In [41], Ghaznavi-Youvalari et al. presented a three-parameter linear function to improve the intra prediction performance. The parameters of this method were obtained based on a mean square Error (MSE) minimization approach from the reference pixels and their locations. This method achieves good efficiency, however, when the pixel to be predicted is far from the reference samples, the prediction performance decreases with the decline in the correlation. To address this problem, we proposed a fusion-based linear prediction method.

## 3. Proposed Method

As mentioned previously, our proposed fusion-based algorithm includes two parts: fusion-based adaptive template matching (mode 67) and fusion-based linear prediction (mode 68). The overall flowchart is depicted in Figure 2. We termed the VVC’s original intra prediction modes (mode 0~mode 66) as the traditional/original modes. When conducting intra prediction, we have to first decide on whether the prediction mode is traditional. If the mode is traditional, the original mode of the VVC is utilized to obtain the prediction pixels. If the mode is not traditional, we then decide on whether the mode is 67 or 68. If it is mode 67, the fusion-based adaptive template matching is employed. If it is mode 68, fusion-based linear intra prediction is used. Some modes can be selected with less distortion, called candidate modes, by rough calculation. Finally, the Rate–Distortion Cost (RDC) is utilized to determine the best mode from these candidate modes. Since the decoder can perform the same operations as the encoder to obtain the prediction and reconstructed values based on the mode number, we do not need to send extra flag bits.

### 3.1. Fusion-Based Adaptive Template Matching

Usually, high correlation and similar textures are prevalent with blocks, and the TMP algorithm can find candidate blocks with the least errors in the reconstructed area to better predict the region with similar textures. Since it searches and compares pixel by pixel, it usually performs well for blocks with similar textures. The process of template matching is shown in Figure 3, in which $T_{r}$ is the reference template and $T_{i}$ is the candidate template. $B_{p}$ is the block to be predicted and $B_{i}$ is the corresponding prediction block of $T_{i}$. $w_{i}$ is the corresponding weighting factor of the candidate block. First, we find the candidate blocks $T_{i}$ through $T_{r}$ in the reconstructed area, and then $B_{i}$ is utilized to obtain $B_{p}$.

Considering that the error between $T_{r}$ and $T_{i}$ has a large influence on the prediction result, we used it as a key basis for weight selection. To balance the compression performance and time complexity, we limited the searching area to 64 × 64 and chose the best four candidate templates through the MSE minimization criteria. The MSE is obtained by:(2)$$\sigma_{i}^{2}=\frac{\sum_{j=0}^{N-1}{\left(T_{r}\left(j\right)-T_{i}\left(j\right)\right)}^{2}}{N}$$
where $\sigma_{i}^{2}$ is the MSE between the reference template $T_{r}$ and *i*-th candidate template $T_{i}$. N is the total number of pixels in $T_{r}$. If $\sigma_{i}^{2}$ is large, it implies that the difference between $T_{r}$ and $T_{i}$ is large, and the corresponding weight is set to be small. If $\sigma_{i}^{2}$ is small, it implies that the candidate template is close to the reference template, and the weight is set to be large. Consequently, the temporary weight can be obtained by introducing a logarithm function as follows:(3)$$\lambda_{i}=\ln\left(1+e^{-\sigma_{i}^{2}}\right)$$
where $\lambda_{i}$ is the temporary weight of the *i*-th candidate template. By normalization, the final weighting factor $w_{i}$ of the *i*-th candidate template is obtained by:(4)$$w_{i}=\frac{\lambda_{i}}{\sum_{j=0}^{3}\lambda_{j}}$$

Then, the final prediction value p(x,y) can be obtained by:(5)$$p\left(x,y\right)=\sum_{i=0}^{3}w_{i}\times B_{i}\left(x,y\right)$$

Generally, the TMP has a high time cost due to the pixel-by-pixel comparison and error calculation. It sacrifices time in exchange for performance gains. Hence, fusion-based adaptive template matching is only employed when the PU size is smaller or equal to 32 × 32. Adding this limitation is important because when the PU size is large, the TMP algorithm will trade high time complexity for coding gain, which is not worthwhile. Simultaneously, the texture information of large PUs is simple, so other modes can achieve good results.

### 3.2. Fusion-Based Linear Prediction

Linear prediction is a simple but efficient method because there is usually a high linear correlation between the pixels to be predicted and the reference samples. However, as the pixels to be predicted move away from the reference pixels, the correlation between them will weaken, and the performance of linear prediction will decline. To address this issue, we present a fusion-based linear prediction method based on Ghaznavi-Youvalari’s work [41].

When the PU size is small, single three-parameter linear prediction can achieve good results. However, when the PU size is large, prediction accuracy declines with the weakening in correlation. Hence, we combined the three-parameter mode with planar mode to obtain the final prediction value. Planar mode obtains the prediction value by weighting the pixels in the horizontal and vertical directions, where the weights are related to the distance. The prediction process of the planar mode is shown in Figure 4.

The prediction value of horizontal $p_{h}\left(x,y\right)$ is obtained by:(6)$$p_{h}\left(x,y\right)=\left(lw-x-1\right)\times b+\left(x+1\right)\times a$$
where (x,y) is the position of the current pixel. a, b are the reference samples and lw is the width of PU. Similarly, the prediction value of vertical $p_{v}\left(x,y\right)$ can be obtained by:(7)$$p_{v}\left(x,y\right)=\left(lh-y-1\right)\times d+\left(y+1\right)\times c$$
where c, d are the reference samples and lh is the height of PU.

Then, the final prediction value is the average of $p_{h}\left(x,y\right)$ and $p_{v}\left(x,y\right)$.

We can see that the prediction process of the planar mode is very close to linear prediction. Hence, the combination of planar mode and three-parameter linear mode can partially compensate for the shortcomings of a single linear prediction and improve the prediction precision.

When the PU size is smaller or equal to 32 × 32, three-parameter linear prediction is used to obtain the final pixels. The linear function is constructed by:(8)$$p\left(x,y\right)=a_{0}\times x+a_{1}\times y+a_{2}$$
where p(x,y) is the prediction value. $a_{0},\;a_{1}$, and $a_{2}$ are the parameters to be calculated. The specific solution is shown in [41].

When the PU size is larger than 32 × 32, both three-parameter linear prediction and planar mode are utilized to obtain the final prediction value:(9)$$p\left(x,y\right)=w_{1}\times p_{linear}\left(x,y\right)+w_{2}\times p_{planar}\left(x,y\right)$$
where $p_{linear}\left(x,y\right)$ is the prediction value of three-parameter linear prediction and $p_{planar}\left(x,y\right)$ is that of the planar mode. $w_{1}$ and $w_{2}$ are the corresponding weighting factors. Since three-parameter linear prediction and planar mode adopt different prediction methods and they can provide different prediction information, they are equally important. Therefore, the two weighting factors were both set as 0.5.

## 4. Experimental Results and Analysis

### 4.1. Experimental Environment

To verify the effectiveness of our proposed fusion-based algorithm, we implemented it on top of VVC Test Model 9.1 (VTM9.1). Coding conditions were followed by the Joint Video Exploration Team (JVET) Common Test Condition (CTC) [42]. Sixteen test sequences with four kinds of resolution were utilized. The resolutions included 416 × 240, 832 × 480, 1280 × 720, and 1920 × 1080. We selected four common test QP values ∈{22,27,32,37} to code video sequences in this work. Only the first 30 frames of each sequence were coded with All Intra (AI) configurations due to the experimental conditions.

### 4.2. Compression Performance

First, we tested the compression performance of our proposed fusion-based algorithm by comparing it with the VVC anchor. The Bjøntegaard Delta Rate (BD-Rate) method [43] was utilized to assess the compression performance. In addition to the respective BD-Rate of the three components, we also calculated the weighted BD-Rate of the three components by:(10)$$BDRate_{YUV}=\frac{\left(4\times BDRate_{Y}+BDRate_{U}+BDRate_{V}\right)}{6}$$
where $BDRate_{YUV}$ is the weighted YUV BD-Rate. $BDRate_{Y}$, $BDRate_{U}$, and $BDRate_{V}$ represent the bitrate of Y, Cb, and Cr, respectively.

Considering that class F is the screen content sequence and the others are the natural content sequence, we made a distinction between them when calculating the average BD-Rate. If the BD-Rate is negative, it indicates that the performance of the VVC has improved. Otherwise, the coding performance of the VVC has deteriorated. As shown in Table 1, compared with the VVC anchor, our proposed fusion-based algorithm saved a bitrate of 0.89%, 0.84%, and 0.90% on average (up to 2.69%, 2.81%, and 2.81%) for components Y, Cb, and Cr, respectively. Simultaneously, our algorithm was particularly efficient for sequences such as “BasketballDrive” (1920 × 1080), “BQTerrace” (1920 × 1080), “Johnny” (1280 × 720), “BasketballPass” (416 × 240), and so on. This is mainly because there are many similar areas in these sequences. The correlation between blocks, and the correlation between reference pixels and the pixels to be predicted were relatively high. For most sequences in Class C and Class D, their texture content was very rich and the texture information varied greatly. For these two classes, the performance of our algorithm was not particularly outstanding since our proposed algorithm is suitable for videos with more similar texture areas. For almost all sequences, our proposed fusion-based algorithm achieved good results, which verified its effectiveness.

Figure 5, Figure 6 and Figure 7 illustrate the Rate–Distortion (RD) curves of some sequences in QP 22, 27, 32, 37. The horizontal axis is the bitrate and the vertical axis is the YUV-PSNR. If the curve of our proposed algorithm is above the VVC, this shows that the peak signal-to-noise ratio (PSNR) of our proposed algorithm is higher than that of VVC for the same bitrate, indicating that our proposed algorithm enhances the compression performance. Otherwise, our proposed algorithm deteriorated the performance. In Figure 6, the blue curve is VTM9.1, and the red one is proposed. We enlarged a small part (the position of the green rectangle) of each curve to show the comparison of results more intuitively. Clearly, our RD curves were higher than that of VVC. For areas with a similar texture, our proposed fusion-based algorithm could better model the relationship between the reference pixels and pixels to be predicted. Hence, for the same PSNR, our algorithm needs less bitrate, which means that our proposed algorithm improves the compression performance of VVC.

Moreover, we compared our proposed fusion-based algorithm with some existing works to further verify the effectiveness of our work. This is detailed in Table 2. In most cases, the performance of our proposed algorithm was superior to the others. In [28], Yoon et al. optimized the CCLM algorithm and achieved good results, although their work only enhanced the channel correlation between the luminance and chroma components and did not consider the correlation of the stronger chroma components themselves. Therefore, overall, our algorithm performed better. In [32], Yoon et al. obtained the mode of the current PU block through the neighboring blocks and achieved good results. However, they only considered the correlation between the current block and its neighbors, not the correlation between the more distant blocks and the correlation between the reference pixels and the pixels to be predicted. Therefore, in general, our proposed algorithm has superior performance. In [26], Schneider et al. obtained the final prediction value through sparse coding and achieved good results, although the dictionary is finite, which limits the performance of their algorithm. Our proposed fusion-based algorithm calculates the corresponding prediction value according to different PUs, which results in improved performance.

### 4.3. Mode Using Probability

Since our new added modes were in competition with the original VVC intra prediction modes, we calculated the using probability of our new modes, as shown in Equation (11). Five sequences were used to conduct this experiment.
(11)$$\eta =\frac{N_{\mathrm{mod}e}}{N_{total}}\times 100\%$$where η is the using probability. $N_{\mathrm{mod}e}$ and $N_{total}$ represent the using numbers of a single mode and all modes, respectively. If η is large, it shows that compared with other modes, our modes are chosen more often, indicating that our algorithm shows superior RD performance.

The using probability of mode 67 (the mode using fusion-based adaptive template matching) and mode 68 (the mode utilizing fusion-based linear prediction) in the luminance component are illustrated in Table 3 and Table 4, respectively.

Both mode 67 and mode 68 were selected to some extent. Specifically, mode 67 was often chosen because of its good performance. Its maximum using probability reached 10.50% for “Johnny” (1280 × 720). The performance of mode 68 was not as appealing as mode 67. Since the TMP algorithm exploits the correlation between blocks and mode 68 exploits the linear relationship between the reference pixels and the pixels to be predicted, TMP generally works better. However, for some blocks, mode 67 could not strike a good balance between complexity and performance, while mode 68 could achieve better results. Hence, mode 68 also made some contributions to video compression. For “BQMall” (832 × 480), the using probability of mode 68 reached 0.28%. These results illustrate that our proposed algorithm showed good compression efficiency.

To show the using probability more intuitively, we plotted the numbers being used in all luma modes of the sequence “BasketballPass” (416 × 240) in QP 32 and “BQTerrace” (1920 × 1080) in QP 27, as shown in Figure 8. The X-axis is the mode number, and the Y-axis is the number being used for each mode. We drew two dotted lines with the numbers being used for mode 67 and mode 68, respectively, to show the comparison of the results more intuitively. Here, the red dotted line represents the numbers of mode 67 being used, and the green one represents the numbers of mode 68 being used. If the blue “+” is below the red dotted line, it means that the using numbers of this mode are less than that of mode 67. Correspondingly, if the blue “+” is below the green dotted line, it means that the numbers being used of this mode are less than that of mode 68. In Figure 8b, there is no red dotted line. This is because mode 67 had been used more than 1200 times and could not be explicitly shown in the figure. Clearly, the numbers being used for mode 67 were much higher than that of most VVC traditional modes. Although the using probability of mode 68 was only 0.29% (“BasketballPass” in QP 32) and 0.22% (“BQTerrace” in QP 22), the numbers being used for mode 68 were still higher than some of the original VVC intra prediction modes.

Table 5 shows the total using probability of mode 67 and mode 68 in the chroma component. For the chroma blocks, the optimal mode can be mode 67 or mode 68, only if that of the corresponding luminance block is mode 67 or mode 68 according to the VVC prediction process. Therefore, the total using probability of our proposed modes in the chroma component was lower than that of the luminance component. In the chroma component, there were also correlations between the different blocks and different modes, hence, our algorithm was still effective for the chroma components.

In addition, we present the modes using subjective graphs of some sequences in QP 27. Figure 9, Figure 10 and Figure 11 show the respective results. The red rectangular boxes are the blocks that utilized our proposed algorithm to conduct intra prediction. As can be seen, in some areas with similar textures, our modes were more competitive. This is because the traditional VVC intra prediction modes are more suitable for blocks with a single texture direction or simple areas, and our proposed algorithm can partly compensate for this shortcoming. These results further verify the good performance of our proposed fusion-based algorithm.

## 5. Conclusions

In this paper, we designed an efficient fusion-based algorithm to enhance the compression performance of VVC intra prediction. First, we presented a fusion-based adaptive template matching method to better predict areas with similar texture. The error between the reference template and candidate template was used to make the prediction results more precise. Second, we presented a fusion-based linear prediction method to fully utilize the correlation between the reference samples and the pixels to be predicted. This method also compensated for the shortcomings of the single linear prediction. Experiments verified the effectiveness of our proposed algorithm. Compared with VTM 9.1, the bitrate savings for components Y, Cb, and Cr achieved 0.89, 0.84, and 0.90% on average, respectively. The maximum bitrate savings were up to 2.69%, 2.81%, and 2.81% for Y, Cb, and Cr, respectively. In addition, the comparison with other works and using probability further verified that our proposed algorithm improved the intra prediction performance. For some classes, the bitrate saving was not that notable. This is mainly because the VVC itself has adopted many new techniques to optimize the performance of intra prediction, which makes it difficult to improve the performance. In the future, we will try to further improve the performance of intra prediction.

## Figures and Tables

**Figure 1 sensors-22-05977-f001:**
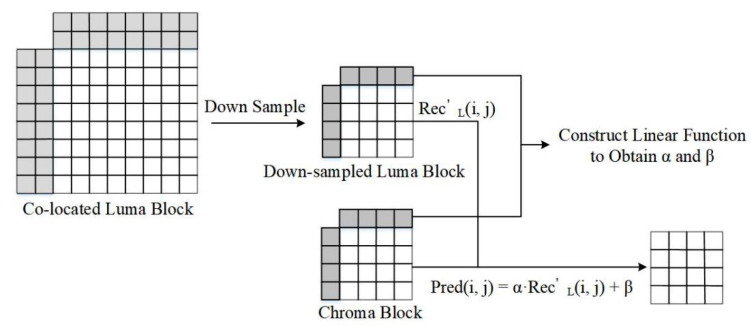
The process of CCLM in VVC.

**Figure 2 sensors-22-05977-f002:**
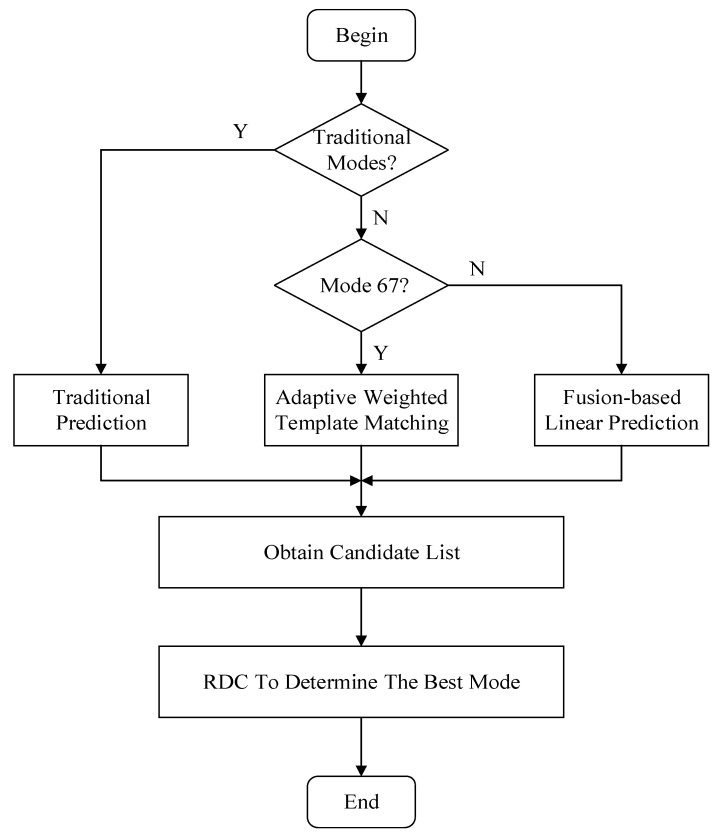
The overall framework of the proposed fusion-based algorithm.

**Figure 3 sensors-22-05977-f003:**
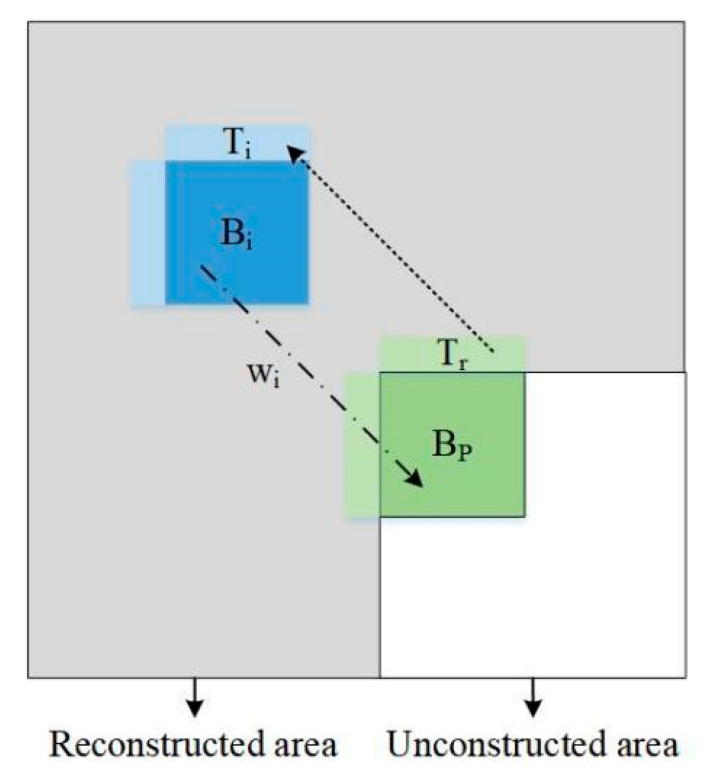
The process of template matching for block $B_{p}$.

**Figure 4 sensors-22-05977-f004:**
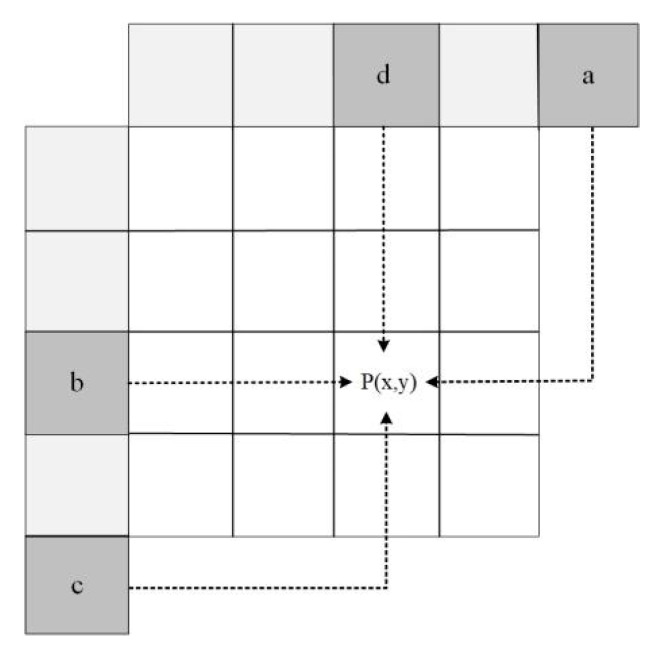
The prediction process of the planar mode based on reference pixels a, b, c, and d.

**Figure 5 sensors-22-05977-f005:**
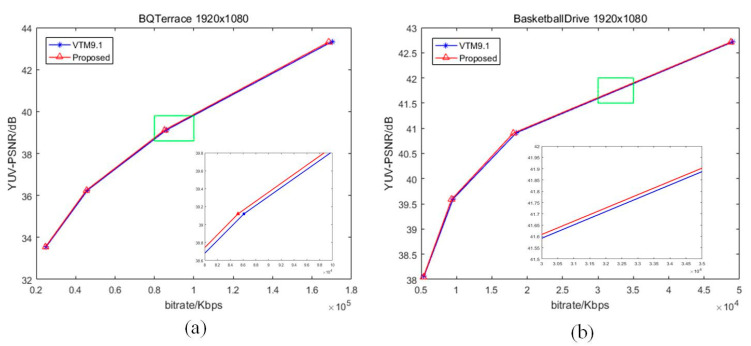
The RD curves of some sequences: (**a**) “BQTerrace” (1920 × 1080); (**b**) “BasketballDrive” (1920 × 1080).

**Figure 6 sensors-22-05977-f006:**
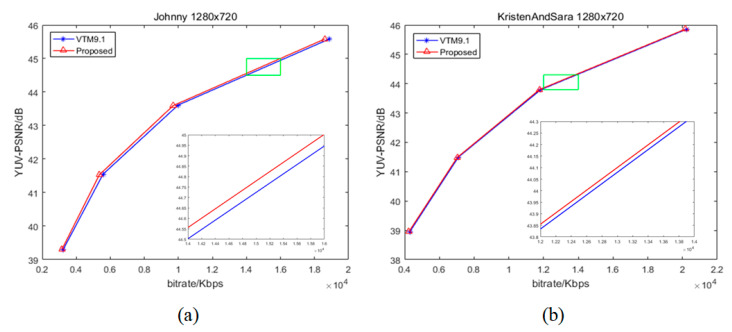
The RD curves of some sequences: (**a**) “Johnny” (1280 × 720); (**b**) “KristenAndSara” (1280 × 720).

**Figure 7 sensors-22-05977-f007:**
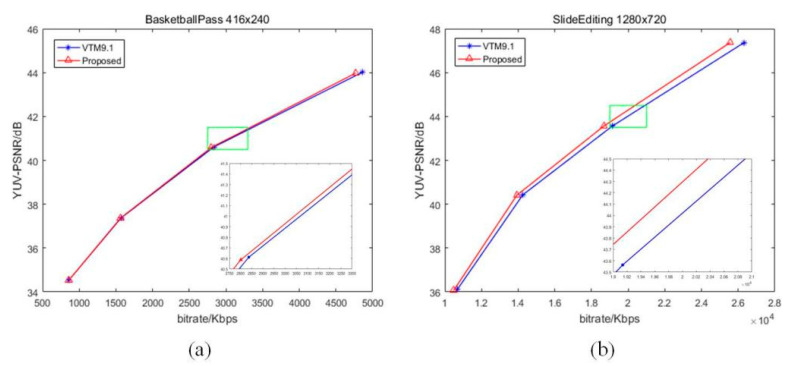
The RD curves of some sequences: (**a**) “BasketballPass” (416 × 240); (**b**) “SlideEditing” (1280 × 720).

**Figure 8 sensors-22-05977-f008:**
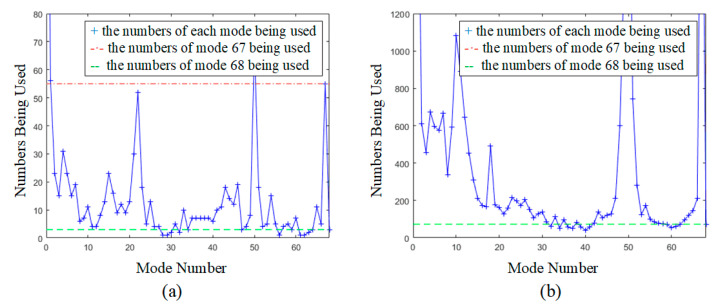
The numbers being used for each mode: (**a**) “BasketballPass” in QP 32; (**b**) “BQTerrace” in QP 22.

**Figure 9 sensors-22-05977-f009:**
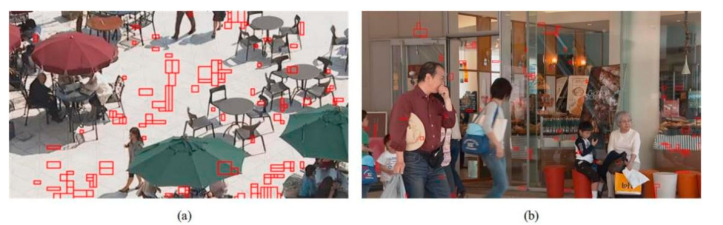
The mode subjective graphs. Images from the JVET CTC [42]: (**a**) “BQSquare” (416 × 240); (**b**) “BQMall” (832 × 480).

**Figure 10 sensors-22-05977-f010:**
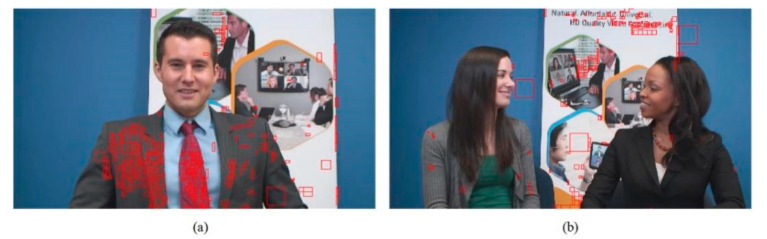
The mode subjective graphs. Images from the JVET CTC [42]: (**a**) “Johnny” (1280 × 720); (**b**) “KristenAndSara” (1280 × 720).

**Figure 11 sensors-22-05977-f011:**
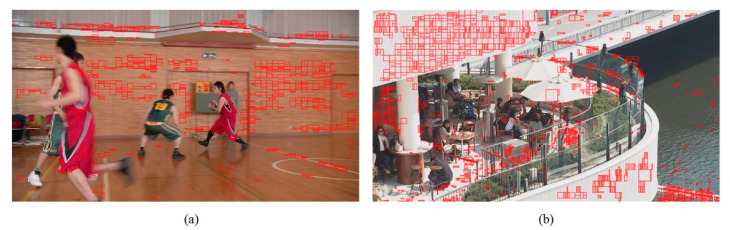
The mode subjective graphs. Images from the JVET CTC [42]: (**a**) “BasketballDrive” (1920 × 1080); (**b**) “BQTerrace” (1920 × 1080).

**Table 1 sensors-22-05977-t001:** The performance of our proposed algorithm compared with the VVC in BD-Rate.

Class	Sequence	Y/%	U/%	V/%	YUV/%
B (1920 × 1080)	BasketballDrive	−1.55	−1.46	−1.62	−1.55
	BQTerrace	−1.08	−1.15	−1.06	−1.08
	Cactus	−0.79	−0.62	−0.95	−0.79
**Average**		** − ** **1.14**	** − ** **1.08**	** − ** **1.21**	** − ** **1.14**
C (832 × 480)	BQMall	−0.25	−0.20	−0.19	−0.23
	BasketballDrill	−0.41	−0.23	−0.64	−0.42
	PartyScene	−0.14	−0.12	−0.35	−0.17
	RaceHorses	0.02	−0.03	−0.31	−0.04
**Average**		** − ** **0.20**	** − ** **0.15**	** − ** **0.37**	** − ** **0.22**
D (416 × 240)	BasketballPass	−0.53	−1.31	−1.50	−0.82
	RaceHorses	−0.01	−0.18	−0.32	−0.09
	BlowingBubbles	0.00	0.17	0.15	0.05
	BQSquare	−0.26	−0.23	−0.47	−0.29
**Average**		** − ** **0.20**	** − ** **0.39**	** − ** **0.54**	** − ** **0.29**
E (1280 × 720)	Johnny	−2.69	−2.81	−2.81	−2.73
	FourPeople	−0.47	−0.55	−0.46	−0.48
	KristenAndSara	−0.81	−0.53	−0.10	−0.65
**Average**		** − ** **1.32**	** − ** **1.30**	** − ** **1.12**	** − ** **1.29**
F (1280 × 720)	SlideShow	−0.72	−1.30	−0.53	−0.79
	SlideEditing	−2.28	−2.32	−2.32	−2.29
	BasketballDrillText	−0.52	−0.40	−0.47	−0.49
	(832 × 480)				
**Average**		**−1.17**	**−1.34**	**−1.11**	**−1.19**
**Overall Average** **(Excluding D And F)**		**−0.89**	** − ** **0.84**	** − ** **0.90**	** − ** **0.88**
**Overall Average** **(Including D And F)**		** − ** **0.81**	** − ** **0.85**	** − ** **0.87**	** − ** **0.82**

**Table 2 sensors-22-05977-t002:** A comparison with other works in BD-Rate.

Class	BD-Rate/%	Yoon et al. [28]	Yoon et al. [32] (Test 3)	Schneider et al. [26]	Ours
**B**	Y	−0.04	−0.45	−0.33	**−1.14**
	U	−0.74	−0.39	-	**−1.08**
	V	−0.83	−0.38	-	**−1.21**
**C**	Y	−0.07	−0.47	−0.2	−0.20
	U	−0.72	**−0.45**	-	−0.15
	V	−0.85	**−0.29**	-	−0.37
**E**	Y	−0.02	−0.58	**−0.3**	**−1.32**
	U	−0.29	−0.45	-	**−1.30**
	V	−0.10	−0.52	-	**−1.12**
**Average**	Y	−0.04	−0.50	**−0.28**	**−0.89**
	U	−0.58	−0.43	-	**−0.84**
	V	−0.59	−0.40	-	**−0.90**

**Table 3 sensors-22-05977-t003:** The using probability of mode 67 in the luminance component.

Sequence	QP = 22	QP = 27	QP = 32	QP = 37	Average
BasketballPass	9.54%	7.48%	5.25%	4.06%	**6.58%**
BQMall	1.62%	1.48%	1.54%	1.44%	**1.52%**
Johnny	10.14%	11.03%	10.99%	9.84%	**10.50%**
BasketballDrive	5.24%	7.01%	9.16%	9.37%	**7.70%**
BQTerrace	8.27%	7.67%	7.20%	6.22%	**7.34%**

**Table 4 sensors-22-05977-t004:** The using probability of mode 68 in the luminance component.

Sequence	QP = 22	QP = 27	QP = 32	QP = 37	Average
BasketballPass	0.00%	0.00%	0.29%	0.28%	**0.14%**
BQMall	0.80%	0.10%	0.12%	0.08%	**0.28%**
Johnny	0.12%	0.13%	0.06%	0.00%	**0.08%**
BasketballDrive	0.08%	0.03%	0.06%	0.04%	**0.05%**
BQTerrace	0.22%	0.08%	0.08%	0.03%	**0.10%**

**Table 5 sensors-22-05977-t005:** The total using probability of mode 67 and mode 68 in the chroma component.

Sequence	QP = 22	QP = 27	QP = 32	QP = 37	Average
BasketballPass	4.81%	2.15%	1.35%	1.31%	**2.41%**
BQMall	0.42%	0.26%	0.36%	0.00%	**0.26%**
Johnny	3.58%	3.41%	2.79%	3.54%	**3.33%**
BasketballDrive	2.99%	3.72%	5.33%	3.89%	**3.98%**
BQTerrace	4.28%	4.65%	4.69%	3.16%	**4.20%**

## Data Availability

Not applicable.

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
