# Peer review of "Fusion-Based Versatile Video Coding Intra Prediction Algorithm with Template Matching and Linear Prediction"

_sensors, 2022, doi:10.3390/s22165977_

Round 1
Reviewer 1 Report
The authors present an algorithm aimed to improve texture reproduction in intra-mode prediction of VVC.
The paper is well written and clear, english form is fair.
Proposed results seem to improve very slightly the performaces of the intra-mde prediction of VVC as is clearly demonstrated on the images taken from the video sequences.
It seems to me that the proposed method deals with the problem of well reproducing the pattern on flat surfaces whose information content is normally negligible.
The improvements obtained with respect to the VVC as is is below 1% in bit rate, that seems to me almost negligible with respect to the quality improvement of the decoded video.
Also, it would be nice to view the video difference of the standard VVC and the proposed improvement with respect to the original video to verify that the enhancement doesn't introduce unwanted patterns.
This paper is in the line of proposals needed to scrape the bottom of the barrel in video coding.
My opinion is to publish the paper as is even if the expected returns from the scientific community might be poor.
Some marginal observations:
1) In equation (9), the planar mode should be referenced to the pixel within the PU with lowest prediction error and not on the PU corner pixel to reduce the error in each prediction
2) In table 1, BDBR not defined
3) Figure 9 can be better explained or results of modes percentages can be reported as time sequences in another figure
Reviewer 2 Report
Refer to attached comments

Reviewer 3 Report
- Abstract: “[…]we proposed a fusion-based linear prediction method which could compensate for the deficiency of single linear prediction.” Why “could”? Why isn’t the sentence always true? Please add a few words on this issue.
- Abstract: “[…] our algorithm also showed superior performance.” In which way? What type of performance?
- The last sentence in the abstract is a bit redundant. If the intra prediction would not be not improve how else would the algorithm improve the performance? If yes, please highlight this other way.
- First paragraph in the introduction section. Please add a few sentences regarding the motivation for researching video compression algorithms and a motivation for researching the proposed approach.
- Lines 68-70, “Although there were … further improvement.” Please rephrase. This is not convincing.
- Please highlight what is the difference between contribution (1) and (2). Isn’t it the same method applied with different inputs, therefore, a single novelty? Add a few details to each so one can understand the difference.
- Please remove contribution (3) as reporting improved results are not a novelty but a requirement. What exactly is the novel idea in computing the results?
- Lines 81-85. Please more these results in a experimental results section. The introduction is not a place to show visual results.
- Figure 2 is blurry. Please update it’s caption with the algorithm’s name.
- The flying equations in lines 121 and 122 does not look good in academic research, it just shows lack of attention to details.
- The Related “works” section is a bit short. Please add a few more paragraphs.
- Figure 3. It’s not clear why such scheme is required. Isn’t it clear that when mode=67 an algorithm is employed and when mode =68 another one is employed? Why two comparison are required? (“Traditional modes?”, “Mode 67?”). Again why do need Figure 3?
- Please highlight how is the extra information regarding the signaling of these mode encoded in VVC. Please comment on the extra bitrate introduced.
- “[…] TMP algorithm can find candidate blocks which is similar to [...]”. Please further check the manuscript for typos.
- Section 3.1, please further highlight the idea behind the proposed method. Why is this approach useful?
- Figure 4 is blurry. Please update the caption with information regarding the image. Is the method applied to each channel?
- Section 3.1. How big is the search area? Add a few sentences regarding the time-complexity.
- Figure 5 is blurry. Please update the caption with information regarding “a”, “b”, “c”, “d”. What are they? Why are this label order was selected?
- Please improve the mathematical notations. “width”? “height”?
- Line 205, there is a times (x) symbol flying inside the sentence.
- Equation (8), please define the “\dot” symbol and add the corresponding punctuation after the equation.
- Line 207 is empty.
- Line 208. “where”- the sentence starts with small capital.
- Similar for equation (9), line 212 and line 213; eq. (6), lines 194 and 195; eq. (7), lines 198 and 199.
- “[…] the weighting factors are set to 0.5 and 0.5 […]”. Please rephrase.
- The manuscript is missing a short discussion regarding the use of the two fusion methods.
- Please improve the mathematical notations. “three-para”?
- Table 1. Way are the results on Class A missing? Why not report also YUV-BDBR?
- The authors need to improve the motivation. The results for classes C, D, and E are almost non-existent.
- “Overall Average (Excluding D And F) ” why there is a need to exclude the results for the D and F classes?
- Table 1, please include a comparison with state-of-the-art methods.
- Please extend Table 2 with more state-of-the-art results.
- Please update the title of Section 4.3. It’s not clear what exactly is studied in this subsection.
- Figure 9, it’s not clear what these number should represent. Maybe showing results using percentages might improve the presentation.
General comments regarding issues that MUST be solved:
- Please improve the manuscript quality! The text must be further polished! There are many sentences written in a “brute” state.
- Please add a comparison with state-of-the-art results.
- The results are not convincing, there improvement is notable only for class B.
Author Response
Please see the attachment.(Our article has been revised by a native English speaker who is one of the authors of this article)

Round 2
Reviewer 3 Report
My comment 13 was not answered. “Please highlight how is the extra information regarding the signaling of these modes encoded in VVC. Please comment on the extra bitrate introduced.” The discussion in Section 4 must be done much earlier. Please provide an explanation on how these two new modes are signaled to the decoder. E.g., In Figure 2, there are two blocks which influence the decision where a “yes” or “no” decision should be signaled to the decoder. How does the algorithm perform this? Please provide this explanation in the proposed method section, section 3, and not later.
Answer to My comment 15. “Since it searches pixel by pixel, it generally works well.” The “[…] it generally works well.” part does not sound good. Please rephrase.
Answer to My comment 16. I’ve provided the comment so that the authors could clarify this in the manuscript. Please update the figure’s caption with this information.
Answer to My comment 18. Similar as above, please update the figure’s caption with this information.
Answer to My comment 24. The authors clearly didn’t understand my comment. I would suggest to ask for help form an English speaker colleague/third party. The structure of the sentence is not ok. You can write “So, the weighting factors are set as $w_1=w_2=0.5.$” or “So, the two weighting factors are both set as 0.5.”
Answer to My Comment 27. Indeed, it can be computed, however, the authors have enough space in the manuscript to clearly present this information so that other paper can CITE your work and compare with the proposed method.
Answer to My Comment 28. Based on this explanation please add a paragraph in the manuscript.
Answer to My Comment 29. Please add a sentence in the results paragraph explaining this decision.
Answer to My Comment 34. Ok, I fully understand this. However, I don’t understand why the manuscript still contains typos and some parts still need further refinement.
